# Reliability-Based Optimization for Energy Refurbishment of a Social Housing Building

**Marco Manzan [1,\*], Giorgio Lupato [1], Amedeo Pezzi [1], Paolo Rosato [1]**  **and Alberto Clarich [2]**

[1] Department of Engineering and Architecture, University of Trieste, via A.Valerio 10, 34127 Trieste, Italy; giorgio.lupato@gmail.com (G.L.); amedeopezzi@hotmail.com (A.P.); paolo.rosato@dia.units.it (P.R.)

[2] ESTECO S.p.A, AREA Science Park, Padriciano 99, 34149 Trieste, Italy; clarich@esteco.com

\* Correspondence: manzan@units.it; Tel.: +39-040-5583506

**Abstract:** This paper investigates the influence of a stochastic variation of both energy and economic parameters in an optimization loop applied to a refurbished social housing building. Usually, energy and economic optimization procedures rely on the results of an underlying numerical deterministic model which influences both energy gains and economic figures. However, an analyst must always face the random variation of input and parameter data. The unknown data can represent poor initial information or data that can change in a long time; this is the case of fuel cost and economic indexes in particular. This paper deals with both problems for building refurbishment optimization, the former related to the initial state of a building, and the latter to the energy cost variability. Reliability analysis considers a stochastic variation of parameters looking for solutions that incorporate a risk level; in this case, it deals with optimization objectives related to different impacts on economic, environmental and health aspects. The considered building represents a social house, and the energy reduction measures involve the application of internal insulation layers to the walls and the replacement of existing windows with more efficient ones.

**Keywords:** optimization; reliability analysis; building refurbishment; discounted cash flow analysis; polynomial chaos

---

## 1. Introduction

Energy consumption in the residential sector in Italy covers 36% of the national final energy use, a large amount, especially if compared to the transport sector that absorbs 32%, and the industrial sector, responsible for a 23% share [1]. Furthermore, the highest share of energy in the residential sector is due to building heating, especially in northern Italy, due to the age of constructions with poor building fabric and insulation characteristics.

Italy is committed to reducing its energy consumption and limiting emissions with an undoubtable benefit to the environment and citizens' health.

Large efforts have been conducted in order to increase plant efficiencies, especially with the substitutions of old boilers with newer condensing ones, and the exploitation of renewable energy sources. To extend the process, important investments should also be focused on refurbishment activities [2]. However, when an operator faces the refurbishment of existing buildings, he faces the problem of large investment costs, which can become a limiting factor; in this case, investors should carefully carry out risk assessment for each intervention [3].

The task is clearly multidisciplinary and involves both accurate energy and economic analysis, in order to define a suitable approach. However, the impact of restructuring measures on the health of inhabitants must also be considered.

In [4], a retrofit proposal was studied which led to a significant reduction in the overall energy demand in a social house in Spain; moreover, the retrofitted building showed an increase in comfort conditions. The overheating risk in refurbished social buildings was described in [5], where 86 rooms in 46 homes were monitored by recording the internal temperatures; the authors highlighted that living rooms with vulnerable occupants showed an increased risk of overheating due to their behavior, confirming the requirement to prevent such situations. In [6], the effect of refurbishment solutions in post-war office buildings was studied; however, even if significant reductions in energy consumption were found, the solutions failed to deliver thermal comfort in summer due to overheating, requiring greater attention in providing summertime cooling measures. Energy simulation techniques are nowadays widespread; they allow for previewing the effect of energy refurbishment efforts and the results are also the base to perform cost evaluation procedures. For example, the beneficial effect of subsidies was highlighted in [7], where the economic feasibilies of different energy efficiency retrofits for social houses are compared. Different energy efficient measures and their economic impact for a heating system were analyzed in [8] using different energy cost escalation rates.

The energy retrofit of a building is a typical case where different solutions characterized by a great number of parameters should be taken into account at the same time. In this case, in the literature, optimization techniques are gaining great interest since they allow one to restrict the possible solutions to an optimal subset based on specified goals.

When dealing with building refurbishment, one of the optimization goals takes into account the economic feasibility of the intervention, prompting users to select solutions with an adequate trade-off between energy and economic prospects. As an example, Ascione in [9] searched for the optimal solutions for building refurbishment taking into account both energy and the costs of the intervention. Lupato [10] highlighted the effect of climatic data on the results of an optimization loop for the refurbishment of a social house, using as objectives the overall energy consumption and the present net value of the investment. A similar approach [11] highlighted the beneficial effect of subsidies.

Usually optimization techniques apply a deterministic approach, fixing some parameters, while changing others during the optimization process. However, while dealing with economic analysis, it is common to incur in situations where parameters are not under control and can vary during the time, especially with long building's lifetime. In order to introduce this approach into an optimization process, uncertainty in searching for optimal designs should be added to the process. This is even more important in a refurbishment process, where an investor requires not only a cost analysis, but also must evaluate the economic risk intrinsic in each investment.

Some authors have pointed out the requirement to analyze the effect of uncertain parameters in building simulation. In [12], a probabilistic method for risk assessment was used in computing the energy requirement and utility cost using a reference commercial building; they computed mean values and standard deviation for identifying the risk associated with a project. In [13], Chary et al. performed a stochastic assessment of the energy performance of buildings considering twelve different regions in Europe, identifying the factors with the greatest impact on energy use. In [14], the authors analyzed the uncertainty propagation of the material properties in energy simulations, emphasizing the problem of a correct assessment of physical properties in existing buildings in order to optimize energy refurbishment. In [15], Di Giuseppe et al. worked on a case study performing a sensitivity analysis and identifying the main parameters affecting the life cycle cost of a building. The research found the financial factors, inflation, discount rate and the energy trend uncertainty as the most influential parameters. Other works are related to the use of different methods for carrying out sensitivity and uncertainty propagation, different methods can be applied as reported in [16], where uncertainty analysis methods were tested to determine an appropriate energy consumption threshold for energy performance contracting. The authors also developed a sort of guideline for applying such techniques in different situations.

While in the literature sensitivity analysis of energy building simulation is a well-established field of research, the problem of the optimization under uncertainty is less studied. A study by Cano et al. [17] applied a stochastic multi-staged optimization algorithm, highlighting the requirement to introduce stochastic variables for risk assessing and decision making. The effect of boundary conditions can affect the results of an optimization; in [18], a form optimization was carried out, taking into account the uncertainties of the parameters; however the uncertainty was not directly entered into the optimization loop, since the authors performed a parametric analysis introducing uncertain quantities. It is interesting to note that, considering the uncertainty, in some cases it could lead to an optimized solution performing worse than the original one.

In this work, a true reliability-based design optimization (RBDO) is carried out; that is, the optimization results are obtained by integrating a genetic algorithm with a stochastic object resulting from stochastic inputs related to economic parameters, therefore the results are not reported as a set of optimized solutions, but instead as solutions that can be obtained with a defined probability given the possible values of stochastic inputs. In particular, the problem considered the refurbishment of an existing building considering also the economic impact of energy conservation measures. Two optimizations are carried out. The former considers two objectives related to the energy performance and economic feasibility of the intervention, and the latter adds an objective related to the comfort conditions during the summer period. For both cases, uncertainties have been embedded into the numerical process in order to develop a methodology to support potential investors in making decisions.

## 2. Building Description

An existing building in Trieste, a city in north-east Italy, whose main features have already been described [11], was chosen to run the reliable optimization. It is composed by four blocks with apartments adjacent to each other. Each block consists of four floors with two small apartments each. The ground floor apartments consist of a kitchen, a bathroom and a bedroom. Each level above the ground floor features two apartments which, respectively, contain one and two bedrooms, a bathroom and a kitchen. The ground level floors and the third level ceilings are adjacent to aerated spaces. The base building was built with massive structures without insulation; the thermal characteristics of external walls have been measured using a heat flow meter, as reported in [19]. The energy refurbishment was carried on by adding insulating layers to the vertical and horizontal internal surfaces in order to preserve the facades. Figure 1 presents the façade of the buildings, while Figure 2 presents the floor plans with five types of flats; Table 1 presents the percentage of characteristic area for each one.

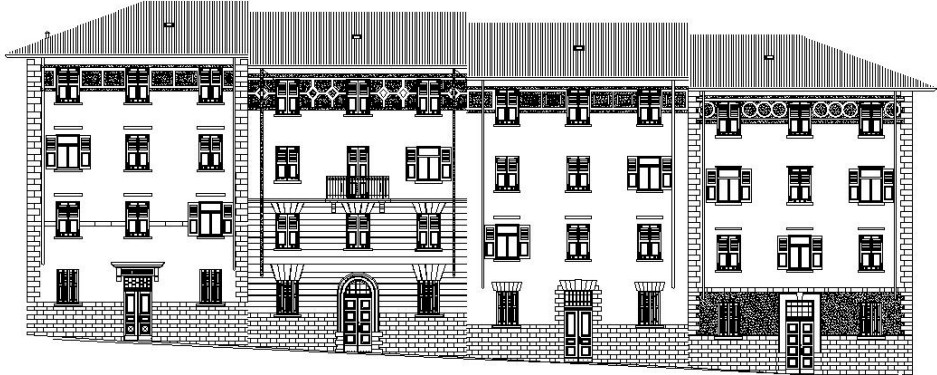

**Figure 1.** Front view of the building.

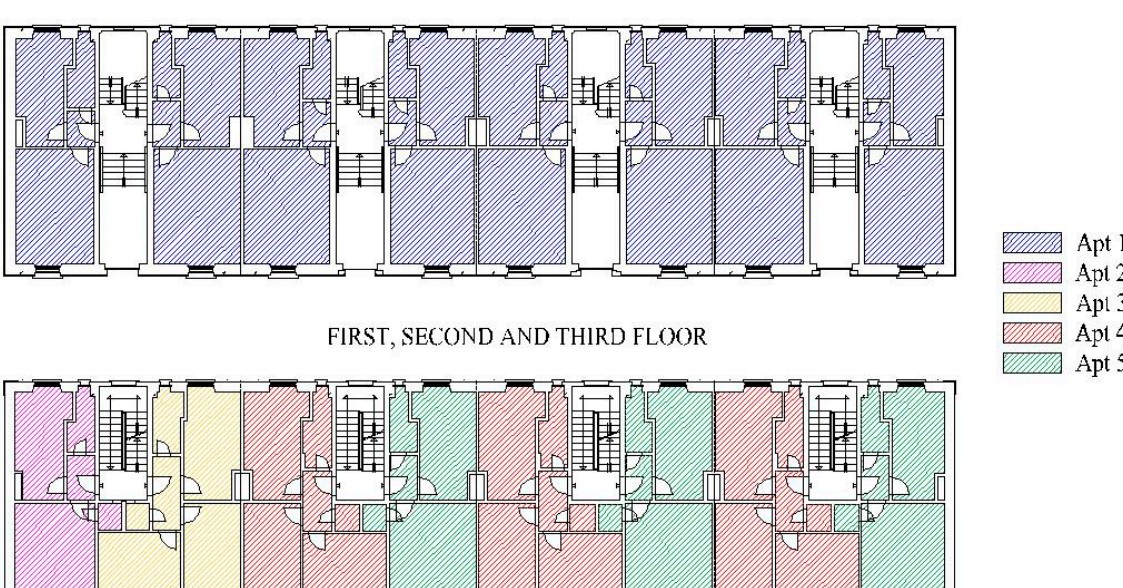

**Figure 2.** Building plants and flat types.

**Table 1.** Distribution of the spaces for each flat.

| Configuration | Living Room + Kitchen | Other Conditioned Spaces |
|---|---|---|
| Apt 1 | 34.00% | 66.00% |
| Apt 2 | 30.00% | 70.00% |
| Apt 3 | 26.00% | 74.00% |
| Apt 4 | 28.00% | 72.00% |
| Apt 5 | 31.00% | 69.00% |

## 3. Building Model Description

The basic model was created using DesignBuilder and the input file in idf EnergyPlus format was saved as the basis for subsequent calculations. During the optimization cycle, the base file was continuously modified according to the required configurations and the calculation phase used EnergyPlus as a simulation engine. It is worth noting that this approach was also allowed to operate in parallel during optimization runs by exploiting the multicore features of the hardware. A simplified approach was used for building's modelling in order to speed up energy computation. Every single apartment was modelled as a unique space while keeping partitions between each other and between apartments and common spaces. However, in order to consider the physical presence of the partitions inside each apartment, equivalent internal masses were added to assure the correct thermal inertia to the system. Internal gains due to people and equipment followed the pattern for residential buildings of EN ISO 13790. The equipment loads were weighted using the percentages of the areas reported in Table 1 obtaining the total internal gain for each type of apartment, as shown in Table 2. The people contribution was inserted into the model considering two metabolic activities: sleeping condition, with 85 W/person and light working, with 110 W/person. The metabolic rates were multiplied by the number of persons, maximum two due to the size of the flats, to obtain people internal gains. These loads were subtracted from the equipment ones, since EN ISO 13790 already considers this contribution. No gains were modelled for entrances, circulation spaces and bathrooms and lighting gains were considered as included in internal gains.

**Table 2.** Distribution of internal gains.

| Day | Hours | Flat Type | | | | |
|---|---|---|---|---|---|---|
| | | 1 | 2 | 3 | 4 | 5 |
| | | W/m$^2$ | | | | |
| Monday | 07:00 to 17:00 | 3.38 | 3.10 | 2.82 | 2.96 | 3.17 |
| - | 17:00 to 23:00 | 7.46 | 6.70 | 5.94 | 6.32 | 6.89 |
| Friday | 23:00 to 07:00 | 4.64 | 4.80 | 4.96 | 4.88 | 4.76 |
| Saturday | 07:00 to 17:00 | 4.04 | 3.80 | 3.56 | 3.68 | 3.86 |
| - | 17:00 to 23:00 | 9.44 | 8.80 | 8.16 | 8.48 | 8.96 |
| Sunday | 23:00 to 07:00 | 4.64 | 4.80 | 4.96 | 4.88 | 4.76 |

Air infiltration was computed in a simplified manner considering an air change rate of 0.50 vol/h for each apartment during winter, while a variable air flow was adopted for summer conditions in order to mimic the opening of windows during summertime, only when external temperatures were lower than internal ones. This approach was implemented in order to account for the impact of refurbishment measurements on dwellers comfort conditions [20]. Windows were considered open when external temperature falls below 2 K of the internal operative temperature. The opening of windows occurs only from May to September and when internal temperatures exceed 24 °C. The WindandStackOpenArea model of EnergyPlus was used to compute ventilation, considering wind effect only.

The heating set-point temperature was 20 °C from 7 a.m. to 2 p.m. and from 4 p.m. to 11 p.m. During the remaining time, a setback temperature of 18 °C was set.

The building blocks had a separate heating plant system which was modelled with an HVAC system composed of a gas boiler and water radiators as terminals in each flat.

Pumps were modelled as variable speed ones. According to Italian law for climatic zone E, the boiler availability was set from the 15th of October until the 15th of April. The heating system water temperature was modelled as modulating through outdoor air temperature sensor.

Circulation spaces and entrances were considered as unheated. Finally, no cooling system was considered, and therefore free floating temperatures were present during the summer season. Domestic hot water was not included into the simulation, and therefore energy consumption takes into account heating energy only.

The weather input file was generated using data recorded between 2001 and 2010 by a weather station 1.6 km away from the building site; with a base temperature of 18 °C, the cooling degree days is equal to 1671. All weather quantities were recorded at 10-minute time intervals, and also comprise measured direct and global radiation avoiding the use of split methods for radiation.

The model could not be calibrated due to the lack energy consumption data. However, the modeling of the baseline building led, for the heating season, to a net energy for space heating of 65 kWh/(m$^2$year). Corrado et al. [21] reported for a similar building located in Milan, with an Heating Degree Day (HDD) value of 2404, a net energy of about 100 kWh/(m$^2$ year), recalibrating this energy value to Trieste climate through the comparison of HDD values, yields a space heating requirement energy of 78 kWh/(m$^2$ year). The calibrated result [22] for a social house in Torino, HDD 2617, reports a net energy need for space heating of 57.6 kWh/(m$^2$ year), corresponding to 46 kWh/(m$^2$ year) for the climate data used in present paper. It appears that the obtained value can be considered representative for the considered type of building. It is worth noting that the difference in the results can be attributed both to the different modeling techniques and to the characteristics of the reference constructions.

## 4. Optimization Approach

The base building external wall is composed by two layers of full-bricks each 25 cm thick. The thermal conductance of the external wall has been measured in [19], while the other characteristics are reported in Table 3 from design projects. The ground floor, the roof and the third level ceilings present a concrete structure whose thickness varies from 15 to 22 cm. External fenestrations consist of

a single-layer glass with high Solar Heat Gain Coefficient (SHGC ) and poor thermal transmittance (Base type in Table 4). Two windows sizes are present; the large size fenestrations with a surface of 1.65 m$^2$ are placed on the north and south walls, the small one 0.262 m$^2$ of surface are present on south wall only.

**Table 3.** Opaque overall thermal transmittances and insulation layers' characteristics and costs.

| Opaque Constr. | $U_{base}$ [W/m$^2$ K] | $U_{ref}$ [W/m$^2$K] | | $t_{ins}$ [cm] | | $C_{ins}$ [€/m$^2$] | | $\lambda_{ins}$ [W/m K] | $\rho_{ins}$ [kg/m$^3$] | $c_{ins}$ [J/kg K] |
|---|---|---|---|---|---|---|---|---|---|---|
| | | *max* | *min* | *min* | *max* | *min* | *max* | | | |
| Wall | 1.55 | 0.822 | 0.215 | 2 | 14 | 9.24 | 40.60 | 0.035 | 25 | 1400 |
| Ceiling | 14.71 | 1.565 | 0.173 | 2 | 25 | 4.93 | 39.61 | 0.036 | 140 | 1030 |
| Roof | 5.88 | 1.35 | 0.170 | 2 | 20 | 13.07 | 71.15 | 0.035 | 25 | 1400 |
| Floor | 2.89 | 1.090 | 0.165 | 2 | 20 | 9.42 | 53.49 | 0.035 | 35 | 1400 |

**Table 4.** Glass solar properties.

| Type of Window | Parameter | Value | Cost € | |
|---|---|---|---|---|
| | | | **Small** | **Large** |
| Base | $U_g$ [W/(m$^2$ K)] | 5.7 | - | - |
| | SHGC [-] | 0.87 | | |
| Type 0 | $U_g$ [W/(m$^2$ K)] | 1.4 | 226.2 | 417.8 |
| | SHGC [-] | 0.66 | | |
| Type 1 | $U_g$ [W/(m$^2$ K)] | 1.2 | 227.3 | 423.2 |
| | SHGC [-] | 0.425 | | |
| Type 2 | $U_g$ [W/(m$^2$ K)] | 0.8 | 244.3 | 500.6 |
| | SHGC [-] | 0.398 | | |

In order to improve the thermal characteristics of the building, a number of refurbishment activities were devised. Building fabric insulation characteristics were improved through the insulation of roof, floors and ceilings that separates the heated areas from crawl spaces. Internal insulation layers were added to the vertical walls, and additionally three types of window were considered to substitute the low performing original ones: double-glass with air gap, double-glass with an argon-filled gap and triple-glass with argon-filled gaps (Type 0, 1 and 2, respectively, in Table 4).

Therefore, the optimization of building refurbishment has been implemented using ten discrete parameters, each representing a different refurbishment action on building elements: seven are related to the opaque surfaces, and in particular four describe the added insulation to vertical walls for each orientation, three for floor, roof, and ceiling, and three are related to windows replacement.

Table 3 reports the lower and higher insulation thicknesses which are changed with discrete values with 1-cm steps. The table also reports the base, lower and higher ranges of overall thermal transmittances for opaque surfaces, along with the insulation material characteristics. The values of the original building are highlighted in grey in the first column.

The economic impact of each solution was computed taking into account the costs of the material of the insulation layers along with the cost of installation. Prices were obtained from the public regional administration price list Prezzario Regionale dei Lavori Pubblici [23], and Table 3 also reports the maximum and minimum cost for each internal insulation intervention. Window prices were acquired from real quotes, adding transport and installation costs—the individual processes for each kind of window are reported in Table 4. Two optimization runs were performed with two and three objectives, respectively. The first run takes into account the economic and environmental impact of refurbishment activities by maximizing the tenth percentile of the net present value (NPV) of the investment and by minimizing the primary energy (PE) consumption of the building. The second

optimization adds as an objective the minimization of the maximum number of hours during which the operative temperature of each apartment is higher than 28 °C ($N_{28}$).

### 4.1. Optimization under Uncertainties – Robust Design and Reliability-Based Design Optimization

Optimization under uncertainties is achieving more and more agreement in the design practice. In fact, most human processes are permeated by uncertainties and this is true especially when dealing with long-term projects involving cost analysis. In energy economics issues, the economic parameters are not fixed, but characterized by some fluctuations that can change the problem outcome and have to be estimated in some manner, usually projecting into the future known past behaviors.

This uncertainty is commonly transferred to the performance of the system, which cannot be determined with an exact and single value, but which is better described by a statistical distribution of results.

In the literature, the main approach to deal with this kind of problems is robust design optimization [24,25], which basically consists in evaluating, for each candidate design proposed by the optimization algorithm, the stochastic distribution of its performances, and in defining objectives based on mean and standard deviations of the same. For instance, it maximizes mean performances and minimizes their standard deviations, in order to optimize the stability at the fluctuations. The strategy is particularly efficient, also because it may take advantage of the polynomial chaos expansion [26,27] regression model, an efficient methodology which exploits proper orthonormal polynomials to analytically estimate with high accuracy the mean and standard deviation, through a reduced number of sampling evaluations. The limitation of this approach is that it normally requires doubling the number of objectives for each performance criteria, having the need to optimize both the mean performance and to minimize its standard deviation, that normally produce an high computational effort to solve the optimization problem.

In this context, another frequent design requirement is the satisfaction of constraints or limits, which should be achieved for a certain percentage of the performance distribution, or for which the percentage of solutions that do not meet the limits or the probability of failure, it must be minimized as much as possible to improve product reliability and quality [24]. The same approach can be extended to the optimization of energy consumption in buildings, where the desired performance must be achieved, but where the design parameters can vary with a statistical distribution.

The main approach followed in the literature to deal with reliability analysis is the one which implements methodologies like FORM or SORM [16,28], which evaluate the failure probability of any candidate design on the basis of its uncertainties distribution and of the given limits to be respected. One limit of this methodology can be represented by the high number of evaluations that may be required by the algorithm to compute the failure probability with accuracy, which makes often practically unfeasible its application to optimization problems.

For these reasons, we adopt in this paper an alternative formulation of a reliability-based optimization problem, introducing a method which conjugates accuracy and reduced number of evaluations.

The methodology takes advantage of the accurate polynomial chaos expansion regression model [26,27] normally applied to robust design problems, to evaluate the complete cumulative distribution function of the performances of the design, from which it is possible to accurately retrieve the percentiles of designs not meeting the prescribed performance (failure probability) or to be maximized/minimized as optimization objectives. In this way, for each performance criterion, it is possible to define a single objective (maximization/minimization of the percentile) instead of two (mean and standard deviation).

In this approach, the evaluation of the performance function, which can generally be time consuming depending on the simulation software applied, is required only to determine the coefficients of the polynomial chaos expansion during the sampling phase.

Once the coefficients are found, it is possible to express the cumulative distribution function (CDF) of any response performance using the polynomial chaos expansion (PCE) directly, which can be considered as a meta-model of the response, practically free in terms of CPU load. Once the CDF is accurately obtained, we can easily retrieve the value corresponding to the needed percentile of the distribution.

*4.2. Introduction to Polynomial Chaos Expansion*

In order to describe in a probabilistic way, the response of a system subjected to uncertainties, one of the most efficient methodologies to be applied is non-intrusive polynomial chaos expansion (PCE) [26,27]. By sampling the input uncertain parameters according to their probabilistic distribution, the PCE regression model, described by Equation (1), allows us to accurately compute the system performance's probabilistic distribution $\Phi$, which is a function of the input variables of the optimization problem ($x$), and of the uncertainties $\xi$, function of a random event $\theta$.

$$\phi(x, \theta) = \sum_{i=0}^{\infty} \phi_i(x) \cdot \psi_i(\xi(\theta)) \tag{1}$$

In Equation (1), the spectral expansion is given by the combination of particular Polynomials $\psi_i$ function of the uncertain variable $\theta$, and which are orthogonal to their corresponding distribution function: in the case of normal distribution, the polynomials are called Hermite polynomials. For practical reasons, the series is normally truncated to a finite $p$ number of terms, which is function of the Polynomial order considered and of the number of uncertain parameters.

The unknown weight functions $\phi$ are computed for each design proposed by the optimization algorithm ($x$ being fixed) by the minimization of the regression error of the function $\Phi$ computed by the sample points, evaluated according to the distribution $\xi$ of the uncertain variables $\theta$.

The accuracy of the regression model is normally evaluated by performance indexes like Leave-one-out R-square, which consists of iteratively leaving one design out of the training set and evaluating the R-square index by training the regression model with the remaining part of the set, and then averaging the results. It can be proved [29] that the convergence rate to the exact momentum of distributions using polynomial chaos regression is exponential with the number of samples, assuring therefore a high accuracy by a low number of sampling points evaluation.

In the application of the method to the problem described in this paper, it was possible to achieve a Leave-one-out R-square index of over 0.99, with only 40 samples for design and a polynomial degree of the third order.

*4.3. Economic Indexes and NPV Computation of Investment Performances*

The evaluation of the economic performances of the proposed technological solutions was carried out using discounted cash flow (DCF) analysis and with reference to the net present value (NPV) of costs and savings generated by the various solutions, discounted at an appropriate rate ($r$).

The costs are essentially due to the investment ($C_0$) necessary to implement the technological solutions considered, while the savings are calculated from the differences ($S_i$) between the current operating costs and those of the obtained applying the technological solutions. Currently, the operating costs refer to energy consumption only. Assuming an evaluation at constant prices, it is necessary to adopt a real discount rate ($r_r$), removing the effect of inflation ($r_i$) from the nominal discount rate ($r_n$), using the following equation:

$$r_r = \frac{r_n - r_i}{1 + r_i} \tag{2}$$

The net present value is given by:

$$NPV = -C_0 + \sum_{i=1}^{n} \frac{S_i}{(1+r_r)^i} \tag{3}$$

Moreover, hypothesizing a constant $S_i$ Equation (3) becomes:

$$NPV = -C_0 + S_i \frac{(1+r_r)^n - 1}{r_r(1+r_r)^n} \tag{4}$$

The economic performances assessment must take into account, in addition to the most likely values of the economic parameters, also their variability and the possible future trends of the main components of operating cost, energy first. Assuming that it is equal to $r_e$, the real annual rate of increase in operating costs, Equation (4) becomes:

$$NPV = -C_0 + S_i \frac{\left(\frac{1+r_r}{1+r_e}\right)^n - 1}{\frac{r_r - r_e}{1+r_e}\left(\frac{1+r_r}{1+r_e}\right)^n} \tag{5}$$

Rearranged as:

$$NPV = -C_0 + S_i \frac{1+r_e}{r_r - r_e}\left[1 - \left(\frac{1+r_e}{1+r_r}\right)^n\right] \tag{6}$$

The simulation of the economic performances of the technological solutions considered was carried out assuming the values shown in Table 5.

**Table 5.** Economic parameters for the simulation.

| Parameter | Value | Unit | Source |
|---|---|---|---|
| Gas (*) | 0.899 | €/m³ | EUROSTAT |
| Electricity (*) | 0.255 | €/kWh | EUROSTAT |
| Inflation rate ($r_i$) (**) | 1.173 | % | Worldwide Inflation Data; [30] |
| Discount rate ($r_n$) | 4.090 | % | Bank of Italy |
| Energy price trend ($r_e$) | 1.59 (s.d. 1.40) | % | Energy Information Administration |

(*) Mean last 10 years prices (constant price 2017) for household consumers all taxes and levies included. (**) Mean last 10 years.

Most of the economic parameters were assumed constant and equal to the average of the values recorded in the last ten years. In order to investigate the influence of stochastic variables, the future trend in the energy price ($r_e$) was considered to variate following a stochastic normal distribution and the investment cost ($C_0$) follows an uniform distribution, since prices often change between the times of design and final construction [15].

The introduction of stochastic input variables obviously implies a stochastic output. This implies the assumption of a "decision rule" with which to deal with the uncertainty induced by the stochastic variables on the NPV.

There are various ways to introduce the effect of uncertainty in a DCF analysis [31]:

- Deterministic DCF where the inputs are fixed and the risk is represented by the incremental risk premium on the risk free discount rate and the output is fixed NPV;
- Deterministic DCF where the inputs are fixed and valued at "certainty equivalent" and the discount rate is risk free and the output is fixed NPV [32];
- Probabilistic DCF where the inputs are distributions and the discount rate is risk free and the output is a NPV distribution.

In this paper, the third solution was adopted and the optimization process requires the assumption of a "confidence limit" in the NPV to be optimized [33]. Assuming a risk-neutral decision maker and a normal distribution of the NPV, the value to be optimized is the average or modal one. Normally, the decision maker is risk-averse and therefore optimizes a NPV with a probability to be overcame greater than 50%. Unfortunately, to our knowledge, there are no studies that have examined this aspect with reference to the energy requalification choices. Therefore, a very cautious attitude was assumed, hypothesizing a reference NPV with a probability of being exceeded by 90%.

### 4.4. Methodology

The inspection of Equation (6) shows that the two stochastic parameters have an effect on the NPV computation only. This suggests a two-step approach for implementing the optimization loop. First the numerical solution with EnergyPlus is performed once per design, then the energy consumption is used to compute the cash flow $S_i$ and, thanks to the polynomial chaos expansion, to generate the VAN distribution from which it is possible to obtain the desired percentiles.

Once the NPV percentiles have been computed, a RBDO can be performed by applying suitable optimization algorithms. modeFRONTIER allows us to operate with nested projects so this capability has been exploited to carry on the optimization: an external project carries on the true optimization step while an inner project performs the polynomial chaos expansion on the economic computation, providing the external one with the 10th percentile value to be used as objective.

With reference to Figure 3, a Python script has been created in order to allow modeFRONTIER to drive the optimization. The Python script implements the "eppy" library using the parameters provided by the optimizer. It modifies the building model characteristics, creating IDF objects, the script than runs the EnergyPlus simulation, and reads the results of a single run providing the optimizer with the Primary Energy PE, the number of exceeding hours in summer period $N_{28}$ and computes the investment cost $C_0$ and the cash flow $S_i$. The second modeFRONTIER project is then invoked, the stochastic economic parameters are generated, according to statistical distributions and the computed investment cost uniform distribution is generated considering a variation of $C_0 \pm 10\%$ of the computed value. Using the generated input statistical distributions, the NPV output distribution is computed using Equation (6), obtaining the required percentile that is transferred to the outer project. Primary energy, the number of exceeding hours $N_{28}$ and NPV 10th percentile are used as optimization objectives by modeFRONTIER external project in order to define new designs. The computation workflow is represented in Figure 3, where a little Gaussian shape identifies the input and output of parameters with stochastic distribution, while the part enclosed by the dotted line identifies the stochastic simulation. The computation of NPV percentiles has been carried using the polynomial chaos implemented in modeFRONTIER. However, it is worth mentioning that as an alternative solution, a Monte Carlo approach could have been implemented, but with a far slower convergence.

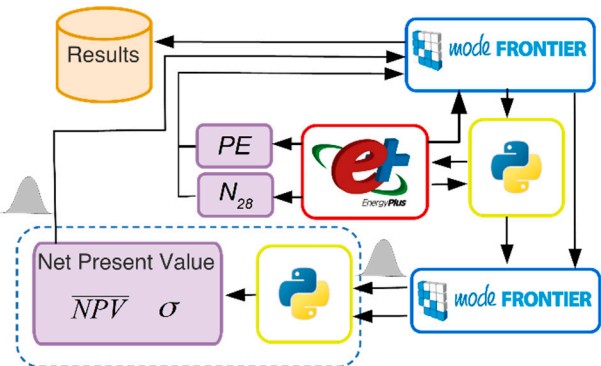

**Figure 3.** RBDO simulation workflow.

*4.5. Optimization Settings*

The optimization considers ten input parameters related to building fabric, two deterministic and one stochastic objectives. Primary energy and the number of exceeding hours are not affected by the stochastic inputs, and therefore can be computed using a deterministic approach. The conversion factors to primary energy were set to 3.167 and 1.084 for electricity and natural gas, respectively, the stochastic output is represented by the 10th percentile of the NPV distributions, or in others words, 90 % of the solution are expected to have a value greater than the objective value. As already pointed out, the choice has been made in order to replicate the decision making of an investor with a low-risk attitude.

The optimization process was performed using the NSGA II optimization algorithm, starting with an initial design of experiments of 24 individuals. The genetic optimization was performed for 50 generations and the optimization lasted 12h 30 min on a 12-core workstation. Since the numerical computation with EnergyPlus and the stochastic optimization were decoupled, the genetic algorithm was the straightforward choice. However, if the stochastic approach was extended to parameters directly affecting the simulation with EnergyPlus, other algorithms, such as response factors or the modeFRONTIER FAST algorithm, would be more appropriate in order to obtain solutions in reasonable time [34].

## 5. Discussion and Results Analysis

Optimizations results are reported using bubble-plots to present up to four variables. The bubble diameters are proportional to the external walls' thermal conductance, south and north oriented, that covers most of the heat losses. Bubble colors represent window types; that is, blue is Type 0, green is Type 1 and red is Type 2. The 10th percentile of NPV and primary energy are the axes of the plot. Since the abscissa of plots is the 10th percentile of NPV, it is worth noting that the reported value means that the 90% of possible solution shows a higher value of NPV, depending on the real values attained by the stochastic parameters.

Figures 4 and 5 refer to optimizations performed using two objectives only, to make the plots clearer they represent the results of only 20 NSGA iterations, it is worth noting as the solutions evolve towards the Pareto front. The inspection of Figure 4 shows that with low energy footprint the solutions are characterized by high insulation levels, as can be inferred by low circles diameter. When the solutions move towards the right of the abscissa axes, the insulation decreases, as can be inferred by the larger diameter of the bubbles. The analysis of window selection is very interesting. In this case, the most performant window is selected only with low energy solutions with low NPV. However, when the solutions evolve towards high NPV values, the optimization selects the less performant one; in this case, the lower investments drive the solutions towards lower insulation levels. Figure 6 presents the comparison between the Pareto frontier of the solutions related to the north and south façades, the figure shows how the optimization selects opaque walls transmittance and drives window selection. Window Type 1 is seldom selected and never for solutions pertaining to the Pareto frontier. For the south wall façade, the solution for windows is always the Type 0, with the highest value of transmittance. On the other hand, on the north wall windows, Type 2 are selected for the low energy case, while Type 0 is the one selected for the high NPV case.

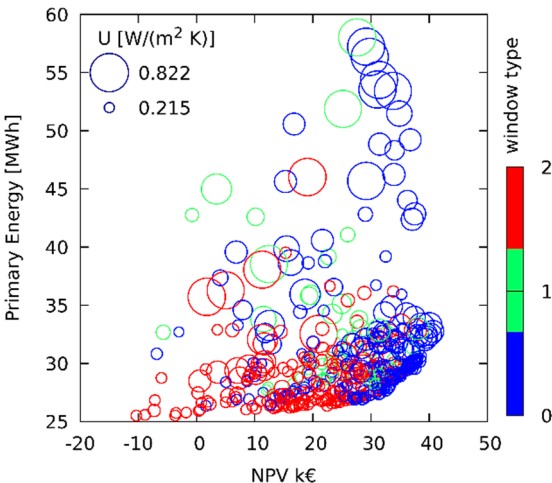

**Figure 4.** Solutions for south wall.

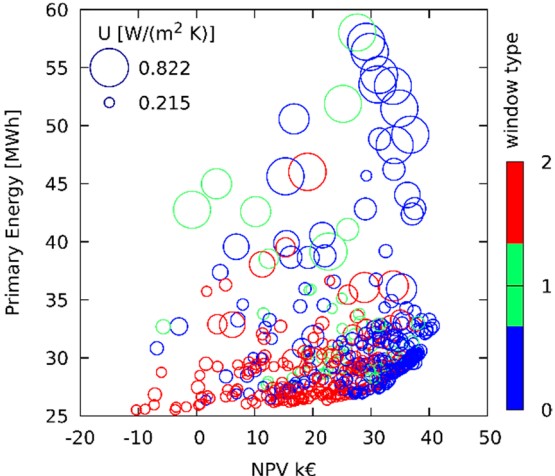

**Figure 5.** Solutions for north wall.

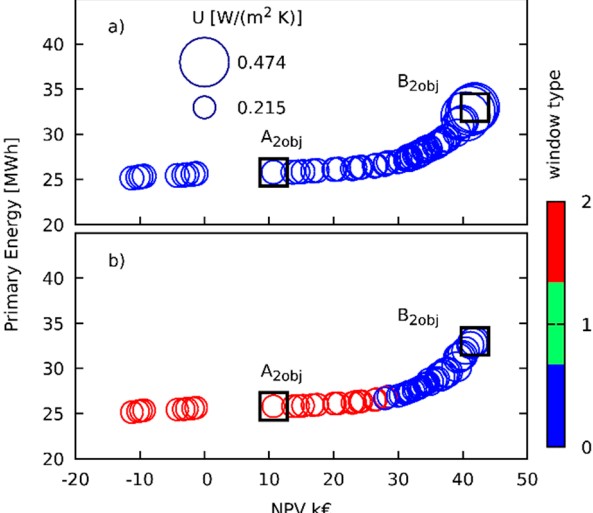

**Figure 6.** Pareto frontier for: (**a**) south wall; (**b**) north wall.

Finally, two designs from the Pareto frontier are compared for each case, they represent the solutions that grant the minimum NPV greater than zero ($A_{2obj}$) and the maximum NPV ($B_{2obj}$). The chosen

designs are highlighted by squares in Figure 6. It can be noted that design $B_{2obj}$ grants the maximum NPV but with a greater energy consumption than $A_{2obj}$. This is due to the fact that $B_{2obj}$ uses low levels of insulation for the south wall and less performant windows for the north one, leading to an increased energy consumption but less expensive refurbishment solutions. The shape of the Pareto frontiers shows as a conflict emerges between the minimization of primary energy consumption and NPV maximization. However, Figure 5 shows that a significant increase in NPV with small increases in primary energy consumption is achievable. Only when the NPV increases above k€ 30 does the primary energy consumption increase significantly. In other words, it is possible to identify technological solutions which at the same time have low energy consumption and good economic performances.

Figure 7 presents the designs for the three objective optimization: the additional objective is represented by the minimization of the maximum number of hours over 28 °C, namely $N_{28}$, for each apartment of the building. In order to highlight the correlation between the parameters and the number of exceeding hours, the designs are categorized as the ones with $N_{28}$ below 380 h, between 380 and 480, and over 480 h. Figure 7a presents the whole set of designs, while Figure 7b presents the solutions on the Pareto Frontier. Figure 8 presents the solutions of the Pareto frontier for the south and north wall using bubble plot representation: as before, the circle diameter is proportional to wall conductance, while the color identifies the window type. However, Figure 8 is difficult to analyze, since no information is given about the $N_{28}$ objective, therefore the designs are separated in Figures 9 and 10 using the aforementioned categories depending on the value attained by the $N_{28}$ objective.

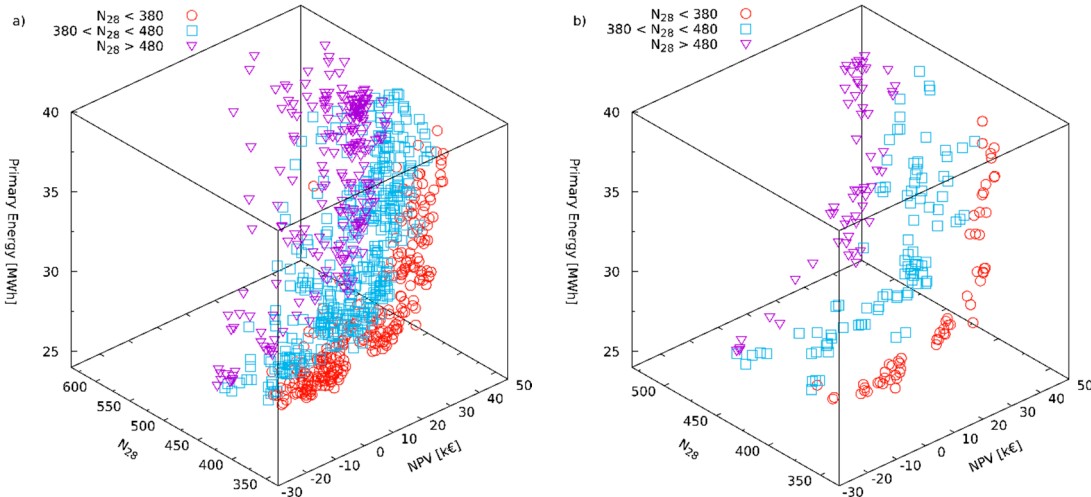

**Figure 7.** Optimization with three objectives: (**a**) all computed designs; (**b**) Pareto designs. The designs are highlighted with different colors for different hours with temperature over 28 °C.

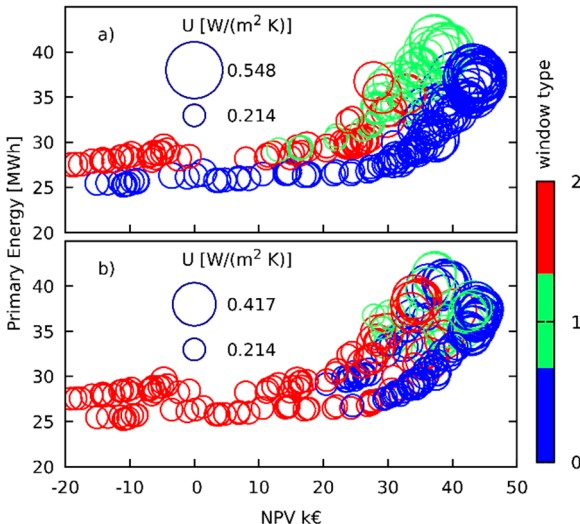

**Figure 8.** Solutions pertaining to the Pareto frontier: (**a**) south wall; (**b**) north wall.

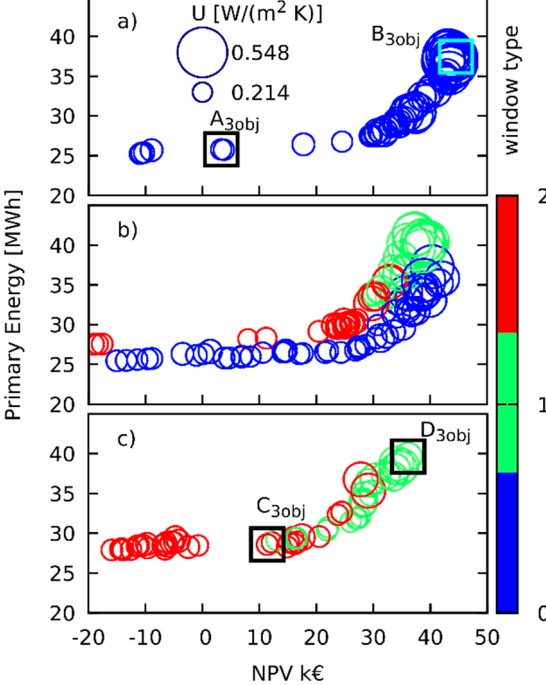

**Figure 9.** Solutions on the Pareto Frontier for South oriented wall with different number of $N_{28}$: (**a**) greater than 480; (**b**) between 380 and 480; (**c**) less than 380.

Figures 9 and 10 present the Pareto frontier of the south and north wall, respectively. It is worth comparing the results with the ones obtained with two objective optimization and reported in Figure 6. For instance, the Pareto frontier of Figures 9a and 10a are quite similar to the results presented in Figure 6a,b, respectively, while on the contrary the results of Figures 9c and 10c show completely different patterns. In order to obtain results with low values of $N_{28}$ the optimization selects solutions with low conductance value for both north and south walls and, above all, windows Type 1 and Type 2 for the south-oriented windows. As a side effect, the energy consumption during the winter heating period increases due to the lower solar heat gain from the windows on the south wall with low SHGC.

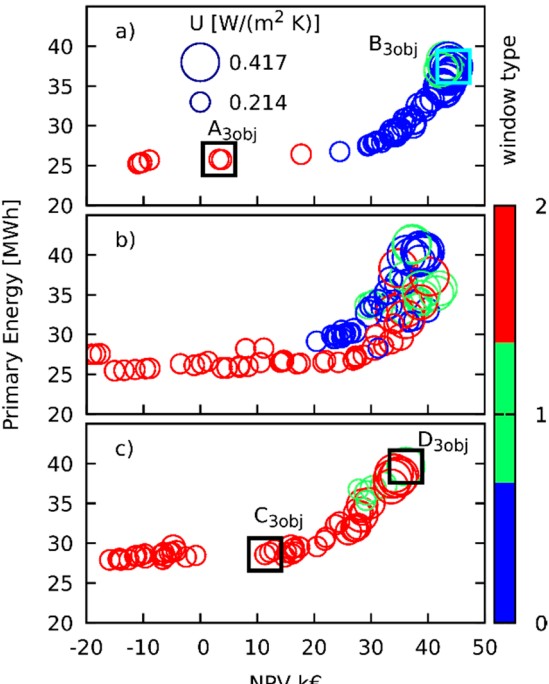

**Figure 10.** Solutions on the Pareto Frontier for north-oriented wall with different number of $N_{28}$: (**a**) greater than 480; (**b**) between 380 and 480; (**c**) less than 380.

It appears clearly that, if it is deemed important to improve the summer internal conditions, reducing the $N_{28}$ objective during summer season, the preferred solutions are the ones less performant regarding the NPV and the PE objectives. These results demonstrate that the multi-objective optimization of energy saving investment on buildings in continental climates, characterized by large seasonal climatic variations, often provides counterintuitive compromise solutions. Therefore, it is very important to correctly weight the importance of each objective of the optimization in order to obtain reliable and efficient performances in the aspects to whom is given more importance.

For a clearer explanation of the different behavior of the designs and how they are influenced by the possible application of the summer performance objective, Table 6 reports six different designs, their performances in terms of PE and NPV and the features of the refurbishment parameters applied. The corresponding solutions are graphically reported in Figures 6, 9 and 10. Two designs, $A_{2obj}$ and $B_{2obj}$, have been selected for the two objective optimization and four designs have been selected for the three objective optimization: two for low PE and with high ($A_{3obj}$) and low ($C_{3obj}$) $N_{28}$ value, other two with the highest NPV, again with high ($B_{3obj}$) and low ($D_{3obj}$) $N_{28}$ value.

By analyzing Table 6, it can be noted that the solutions of the two objective analysis, $A_{2obj}$ and $B_{2obj}$, are similar to those of the three objective with high values of $N_{28}$ $A_{3obj}$ and $B_{3obj}$. Instead the selected designs with low $N_{28}$ values, namely $C_{3obj}$ and $D_{3obj}$, are characterized by higher levels of insulation on opaque walls and, for instance Type 0 window are never selected in this case, and also for the high NPV case Type 1 and Type 2 windows are selected when, for the $B_{2obj}$ and $B_{3obj}$ only Type 0 windows is the chosen one.

**Table 6.** Selected designs.

| Parameter\Id | | $A_{2obj}$ | $B_{2obj}$ | $A_{3obj}$ | $B_{3obj}$ | $C_{3obj}$ | $D_{3obj}$ |
|---|---|---|---|---|---|---|---|
| $U_{wall,S}$ | [W/(m$^2$·K)] | 0.230 | 0.470 | 0.230 | 0.470 | 0.230 | 0.310 |
| $U_{wall,N}$ | [W/(m$^2$·K)] | 0.210 | 0.280 | 0.210 | 0.420 | 0.210 | 0.420 |
| $U_{wall,E}$ | [W/(m$^2$·K)] | 0.240 | 0.280 | 0.280 | 0.280 | 0.240 | 0.280 |
| $U_{wall,W}$ | [W/(m$^2$·K)] | 0.210 | 0.420 | 0.210 | 0.420 | 0.230 | 0.370 |
| $U_{ceiling}$ | [W/(m$^2$·K)] | 0.150 | 0.200 | 0.140 | 0.390 | 0.140 | 0.390 |
| $U_{roof}$ | [W/(m$^2$·K)] | 0.410 | 1.350 | 0.330 | 1.350 | 0.760 | 1.350 |
| $U_{floor}$ | [W/(m$^2$·K)] | 0.190 | 0.490 | 0.150 | 0.490 | 0.230 | 0.490 |
| $Window_{1,N}$ | | Type2 | Type0 | Type2 | Type0 | Type2 | Type1 |
| $Window_{1,S}$ | | Type0 | Type0 | Type0 | Type0 | Type2 | Type1 |
| $Window_{2,S}$ | | Type0 | Type0 | Type0 | Type0 | Type2 | Type1 |
| PE | [MWh] | 25.80 | 32.98 | 25.80 | 37.45 | 28.60 | 39.64 |
| NPV | [k€] | 10.69 | 41.83 | 3.29 | 44.54 | 11.38 | 36.11 |
| $N_{28}$ | [hours] | 466 | 528 | 481 | 519 | 353 | 377 |

The results highlight three different approaches a decision maker can have regarding the problem of building refurbishment. The energy-aware approach implies that the solution with the higher insulation and higher cost are chosen; for instance, this means the triple glazing windows and the higher insulation thickness. However, in this case, the return of the investment can be poor and, in some cases, depending on the economic stochastic parameters, there is a risk to obtain negative NPV, so the energy savings are not sufficient to compensate the initial costs—the solutions identified as $A_{2ibj}$ and $A_{3obj}$ follow this pattern. Another approach aims at having the highest NPV possible, so the reduction in energy consumption is less important; this approach means that the less expensive solutions are the preferred, so solutions such as $B_{2obj}$ and $B_{3obj}$ are selected. However, the results may be excluded due to the regulation requirements which fix minimum threshold values for energy performance—in this case, the limits can be incorporated into the optimization process by introducing suitable constraints. A different approach can also take into account the effect of insulation on summer conditions, which leads to a change in preferred solutions, since a lower value of overheating hours drives the solutions towards higher energy consumption, such as $C_{3obj}$ or a lower NPV as $D_{3obj}$. However, irrespective of the preferred solution, a decision maker using the presented approach has the feeling of the economic risk involved in carrying out the refurbishment. For each solution, there is an high probability to have an economic return of the invested money, for instance the optimization could have used higher values of NPV percentiles (less risk averse decision maker) resulting in higher expected returns but with a lower chance. In fact, the convenience of an energy saving investment depends on climate, economic and technological uncertainties. It follows that the choice of an investment affects both the expected NPV and the probability of reaching it. In our case study, a high risk-adverse decision-maker has been hypothesized and therefore the proposed efficient solutions are very prudent in economic terms.

## 6. Conclusions

Building energy reliability-based design optimization for a social building energy refurbishment has been carried out. Uncertainties on economic parameters have been taken into account by assuming a stochastic distribution of the increase in energy prices during building's lifetime and the investment cost. The objectives of the optimization reflect the stochastic nature of the problem by maximizing the NPV with a probability of being exceeded by 90% in order to reflect the choice of a prudential decision-maker. Two optimizations have been considered, the former using two objectives related to primary energy consumption PE and net present value NPV, the latter adding a third objective in order to search solutions able to minimize overheating problems during summer season. The optimizations led toward solutions with different choices between north and south facades, and furthermore the introduction of an additional objective gave rise to a set of solutions quite different from the ones obtained with only energy and economical objectives, leading to the use of

more performant windows, irrespective of the wall orientation. This represents an important outcome, since economic or energy-related objectives are not sufficient parameters to be taken into account when dealing with building refurbishment. The results also show that the solutions can be variegated and depend on the level of acceptable risk. In this paper, a 10% risk has been selected, meaning 90% of the solutions may give NPV values greater than the ones computed. However, other values can be chosen depending on the amount of risk a possible investor can accept. The use of polynomial chaos expansion in evaluating the stochastic functions allows us, with few computations for each design, to determine the percentiles to be used in the optimization, allowing an easy extension to problems which are more demanding in terms of computational resources.

**Author Contributions:** Conceptualization, M.M. and P.R.; Methodology, G.L. and A.C.; Software, G.L. and A.P.; Validation, M.M. and A.P.; Formal Analysis, P.R. and A.C.; Investigation, A.P., G.L.; Resources, M.M.; Data Curation, A.P. and G.L.; Writing Original Draft Preparation, M.M., P.R., A.C., and A.P.; Writing Review & Editing, M.M. and A.P.; Visualization, M.M. and A.P.; Supervision, M.M.; Project Administration, M.M.; Funding Acquisition, M.M. All authors have read and agreed to the published version of the manuscript.

**Funding:** This research received no external funding.

**Acknowledgments:** The authors wish to thank the ATER (Azienda Territoriale per l'Edilizia Residenziale di Trieste) for the valuable information.

**Conflicts of Interest:** The authors declare no conflict of interest.

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
