# Peer review of "Reliability-Based Optimization for Energy Refurbishment of a Social Housing Building"

_energies, doi:10.3390/en13092310_

Round 1

Reviewer 1 Report

The article „ Reliability-based optimization for energy refurbishment of a social housing building” aims to investigate an optimization strategy for a social house. In addition, the authors carry out a reliability analysis based on Polynomial Chaos Expansion. The research has good potential and it is based on EnergyPlus simulations, which are processed using modeFRONTIER platform.

The English level is good. The research is fairly documented. The bibliographic references are appropriate for this research, most of the cited sources in the article are less than 5 years, showing clearly that there are current preoccupations for the proposed study.

My suggestions to the authors are the following:

The authors have provided in the chapter entitled “Economic indexes and NPV computation of investment performances” the methodology for the economic performance of the building but the evaluation/conclusions were presented just as a part of RBDO simulation workflow without tangible results. Please improve.

Please improve the methodology of the research. It is unclear how the reliability analysis is performed. Just the phrase “In this case some input variables are considered not as fixed values but following a probabilistic distribution.” is ambiguous and can raise many questions.

The conclusions must be improved. The authors present rather some general aspects, without drawing some clear findings of the research.

Reviewer 2 Report

Is it typical to have so many keywords?

How is the model validated?

This study looks like a case design, how to make the findings more general?

The novelty of this study should be further clarified in the introduction.

The abstract and the conclusions should include some quantitative results and some suggestive take-aways.

Reviewer 3 Report

The manuscript analyses the impacts of stochastic variation in data used to optimize parameters involved in building energy retrofitting using social housing in Italy as case study. This considers both low-quality initial data and time variation of input parameters in terms of energy costs. Overall, the study focuses on an important gap in building energy simulation and uses adequate methods to propose an approach to address it. The results are interesting and relevant to work in the area; however, the manuscript needs rework including clarification of details that are important to understand the implications of the results. One key point that should be noted is the lack of a validation procedure that could help quantifying the advantages of the proposed method. The manuscript is clear and well written, and the results are worth publishing.

Section comments:

Introduction. The introduction of the manuscript starts with an overview of the Italian case. Although this is adequate, a more comprehensive overview of the context of the problem based on the energy retrofitting literature would help to make a more definite point regarding the significance of the manuscript. This section reviews critical works related to the impacts of different variables in building simulation, including an overview of sensitivity analysis as a means to deal with them. Then, the authors state the uncertainty in simulation is a less studied issue and present an article dealing with this problem. If there are no more studies dealing with uncertainty in simulation, this should be clearly stated, and the manuscript should be reviewed in detail. Otherwise, given the focus of the manuscript, any other related study should be included in this review. In parallel, if the studies in the area are limited, when possible, the authors may want to include works dealing with similar issues but under different scenarios. There are a number of statements at the beginning of the manuscript that are not supported by references; please add (lines 28 to 43).

Building description. This section helps to understand the case study parameters and assumptions. The building has been studied in the past as clear from the references cited. A more detailed account of the variables involved would help, including photographs (instead of a screen capture of the model) and layout plans, cross-sections and envelope insulation details if possible. Table 1 is difficult to interpret. The original envelope components, HVAC systems and household profiles should be described in detail. The manuscript should include a description of the climate where the building is located, please include in this or other section.

Building model description. This section presents all the modelling parameters clearly and with enough detail. Please state the ach used for simulations in the summer period. Please clarify, in this section, what software was used for the simulations. Based on Figure 1, one could assume that this was done in DesignBuilder, but then Section 4 describes the workflow as a direct Python/eppy interface with EnergyPlus (and modeFONTIER). Please clarify how was the initial idf file generated (include any assumptions), and what epw file was used. Placing Figure 1 and Table 1 in this section may help in terms of readability.

Optimization approach. Please indicate the thermal properties of the envelope walls (as measured). Also, provide a detailed account of the different parameters involved in the optimization, including, if possible, the materials and specific costs considered for envelope wall refurbishment scenarios. The description of the optimization procedures is detailed and precise. One of the main contributions of the manuscript is the proposed method combining Reliability analysis and Poynomial Chaos Expansion. Please clarify how the proposed method is expected to overcome the limitations of these types of assessment without accuracy problems.

Discussion and results analysis. This section presents the results of the optimization using bubble plots. Unfortunately, some of the charts are difficult to read and need adjustment. In Figures 3 and 4, for example, agglomeration in the bottom right prevents reading the size of the circles. This is an important problem as diameter indicates the thermal properties of envelope solutions. Please do not report results that contain mistakes (lines 340-341). Other than that, the results are clearly presented and discussed in enough detail. Nonetheless, this section should rely on the literature presented in the Introduction section to connect these results with previous studies in order to clarify its contribution.

Conclusions. This section summarises the results and main outcomes of the manuscript. Please do not present new data (i.e., selected designs) in this section. Any findings should be presented in previous sections.

Other comments:

The manuscript is clear and well written, but please ensure that all statements are supported by either data or references. Figures and Tables should always be placed in the document after mentioned in the text (e.g., Table 2). For readability, please present all acronyms in the text.

Round 2

Reviewer 1 Report

The revised version addresses the previous comments. Regarding chapter “4. Optimization approach”
I have a few comments:
• on the aesthetic side, I would suggest the authors to move table 3 and table 4 close to the reference point in the text. 
• Sections 4.1, 4.2, 4.3 must be improved. Some of the information is redundant. The performance criteria of the probabilistic model are not provided nor the approach. It is stated just that “In this approach the optimization objective is represented by a predefined percentile to be maximized or minimized”.

Author Response

lease see the attachment.

Reviewer 2 Report

The model has to be validated or verified in some way. Not necessary to use the data for the chosen building, but the method used needs to be verified with enough accuracy. For example, you can use data from the literature, or you can clarify that this method has been well used in other work.

Reviewer 3 Report

The revised version of the manuscript addresses all the comments and resolves/clarifies the main issues identified in the first round of reviews. Please note that the text “A base building model has been created using DesignBuilder, however, EnergyPlus input file has been generated and used to carry on for the optimization run” can be ambiguous. The manuscript should make clear that the EnergyPlus input file (idf) was generated using DesignBuilder, and that this software was not used to conduct the optimisation runs. Interesting research topic and results, thanks.

Round 3

Reviewer 1 Report

The have addresses the previous comments.

Reviewer 2 Report

I have no additional comments.